# Macrophages are necessary for epimorphic regeneration in African spiny mice

Jennifer Simkin[1,2,3], Thomas R Gawriluk[1], John C Gensel[2,3], Ashley W Seifert[1]*

[1]Department of Biology, University of Kentucky, Lexington, United States;
[2]Department of Physiology, University of Kentucky, Lexington, United States;
[3]Spinal Cord and Brain Injury Research Center, University of Kentucky, Lexington, United States

**Abstract** How the immune system affects tissue regeneration is not well understood. In this study, we used an emerging mammalian model of epimorphic regeneration, the African spiny mouse, to examine cell-based inflammation and tested the hypothesis that macrophages are necessary for regeneration. By directly comparing inflammatory cell activation in a 4 mm ear injury during regeneration (*Acomys cahirinus*) and scarring (*Mus musculus*), we found that both species exhibited an acute inflammatory response, with scarring characterized by stronger myeloperoxidase activity. In contrast, ROS production was stronger and more persistent during regeneration. By depleting macrophages during injury, we demonstrate a functional requirement for these cells to stimulate regeneration. Importantly, the spatial distribution of activated macrophage subtypes was unique during regeneration with pro-inflammatory macrophages failing to infiltrate the regeneration blastema. Together, our results demonstrate an essential role for inflammatory cells to regulate a regenerative response.

**\*For correspondence:** awseifert@uky.edu

**Competing interests:** The authors declare that no competing interests exist.

## Introduction

Over the past three decades, regenerative biology has merged its rich historical practice with new genetic tools to discover how animals are capable of regenerating tissue and organs. Regeneration biologists commonly investigate organ regeneration in a range of metazoans including hydra, planarians, crickets, zebrafish, salamanders, newts, lizards and even some mammals. Conventional studies perturb cellular functions or genetic pathways to inhibit the normal regenerative response and thus seek to identify key cellular and molecular mechanisms underlying the regenerative response to injury. Alternatively, some investigators have employed a comparative approach to discover key mechanisms underlying regeneration. In this framework, two related species undergo different responses to injury in identical tissues and exhibit either a regenerative or a scarring response (*Gawriluk et al., 2016*; *Sánchez Alvarado, 2000*; *Sikes and Bely, 2010*; *Wagner and Misof, 1992*). This comparative approach may be particularly useful for unraveling complex interactions such as how inflammation and immunity permit, instruct or inhibit local cells to initiate and undergo functional regeneration in lieu of scarring.

Exactly how cellular inflammation and immunity affect regeneration remains controversial. One perspective posits that inflammation impedes regeneration (*Harty et al., 2003*; *Mescher et al., 2017*), a view supported by reports of less robust immune responses in animals and tissue that regenerate when compared to those that cannot (*Brant et al., 2016*; *Mak et al., 2009*; *Mescher et al., 2013*; *Redd et al., 2004*). Similarly, chronic inflammation leads to compromised healing and fibrotic disease (*Martin and Leibovich, 2005*; *Riches, 1988*; *Wynn, 2004*). However,

**eLife digest** The cells of the immune system are essential to defend an organism from disease. In addition, some of them are also thought to play an important role in helping injured tissues heal or even regrow. For example, when an animal is injured, immune cells such as macrophages rush to the wounded site to clear debris and help repair the damage. Macrophages come in different forms and subtypes, and express different protein markers on their surface, depending on where in the body they reside.

Few mammals can completely renew or regrow a damaged tissue – a process known as tissue regeneration. Instead, humans and most other mammals repair injuries by producing scar tissue, which has different properties compared to the original tissue it replaces. One exception is the African spiny mouse, which, unlike other rodents studied, can regrow skin and fur, nerves, muscles, and even cartilage. It has been shown that in highly regenerative animals such as salamanders and zebrafish, macrophages are necessary to initiate tissue regeneration. Documented cases of tissue regeneration in mammals are rare and therefore less understood. Until now, it was not clear why two species as closely related as spiny mice and house mice would heal identical injuries in different ways.

To better understand how new tissue regenerates, Simkin et al. compared the healing abilities of spiny mice and house mice after they received an injury to their ear and showed that macrophages appeared to be important for both the regeneration of new tissue and the formation of scar tissue. When Simkin et al. removed all macrophages in the ear of spiny mice, their ear tissue could not heal and regrow. When the macrophages were allowed to re-invade the injured site, the tissue in the ear regenerated. Further experiments showed that during tissue regeneration and scarring, different subtypes of macrophages appeared to be active.

The findings suggest that specific subtypes of macrophages could be a key element in helping tissue to regenerate. An important next step will be to further explore the different types of macrophages and whether the injury site determines what types of cells are active. A deeper understanding of how tissues can regrow in mammals will be essential to advancing our ability to stimulate tissue regeneration in humans.

physical injury elicits inflammation during regeneration *and* scarring. Specifically, cytokines and chemokines produced by neutrophils, macrophages and T-cells recruit fibroblasts, promote granulation tissue formation, activate myofibroblasts, and promote collagen production and deposition (*Aliprantis et al., 2007*; *Lakos et al., 2006*; *Mori et al., 2008*; *Ong et al., 1999*; *Smith et al., 1995*). Dampening the inflammatory response by depleting leukocytes creates better healing outcomes following damage to skin, skeletal muscle, and liver (*Dovi et al., 2003*; *Duffield et al., 2005*; *Martin et al., 2003*; *Novak et al., 2014*). Thus, when one considers that injury-mediated inflammation and immunity is an ancient process shared by animals (and plants) that can and cannot regenerate, a more nuanced relationship between regeneration and immunity emerges.

Mounting evidence suggests that certain immune cells may be necessary to induce and sustain regeneration. Depletion of phagocytic cells (e.g. macrophages and dendritic cells) inhibits regeneration in axolotl limbs, zebrafish fins, and neonatal mouse hearts (*Aurora et al., 2014*; *Godwin et al., 2013*; *Petrie et al., 2014*). Furthermore, the timing of leukocyte depletion has a major impact on regenerative outcomes (*Arnold et al., 2007*; *Duffield et al., 2005*; *Varga et al., 2016*) supporting an important role for changing immune cell phenotypes (*Gensel and Zhang, 2015*; *Koh and DiPietro, 2011*; *Mantovani et al., 2013*). Although these findings support a positive function of certain immune cells on regeneration, they also simplify important differences across species. For instance, salamanders lack important T-cell phenotypes and utilize primarily IgM rather than IgG antibodies while mounting an adaptive immune response (*Chen and Robert, 2011*; *Cotter et al., 2008*). While this diversity is of interest to biologists, it may obscure the goal of regenerative medicine – to induce regeneration in humans. This makes mammalian models of tissue regeneration especially relevant to questions regarding what role immune cells play during regeneration.

Since first described by Markelova (cited in *Vorontsova and Liosner, 1960*), ear pinna regeneration has remained an interesting example of musculoskeletal regeneration in mammals (*Gawriluk et al., 2016*; *Goss and Grimes, 1975*; *Joseph and Dyson, 1966*; *Matias Santos et al., 2016*; *Seifert et al., 2012a*; *Williams-Boyce and Daniel, 1980*). Recent work in African spiny mice species (*Acomys cahirinus, A. kempi and A. percivali*) supports ear pinna regeneration as an epimorphic process (*Gawriluk et al., 2016*) aligning it with appendage regeneration in other vertebrate regenerators such as salamanders, newts, zebrafish and lizards. Importantly, not all mammals can regenerate ear tissue providing variation to compare regeneration and scarring in identical tissue (*Gawriluk et al., 2016*; *Williams-Boyce and Daniel, 1986*). Spiny mice are able to regenerate full-thickness skin, blood vessels, nerves, cartilage, adipose tissue and some muscle, whereas the same injury in *Mus musculus* (outbred and inbred strains) leads to incomplete ear hole closure and scar formation (*Gawriluk et al., 2016*; *Matias Santos et al., 2016*; *Seifert et al., 2012a*). Here, we report how the two main orchestrators of inflammation, neutrophils and macrophages, respond to injury during epimorphic regeneration in *Acomys cahirinus* compared to scarring in *Mus musculus*. *Acomys* and *Mus* exhibit the same circulating leukocyte profiles, and we demonstrate a robust acute inflammatory response in both species. We demonstrate higher neutrophil activity in the scarring system compared to higher ROS activity in the regenerative system. We show that macrophages between the two species display similar *in vitro* properties providing a comparable baseline prior to and following injury. We also observed distinct differences in the spatiotemporal distribution of macrophage subtypes during regeneration and scarring. Finally, depletion of macrophages, prior to and during injury, inhibited blastema formation and regeneration, thus demonstrating a necessity for these cells.

## Results

### Circulating leukocyte profiles are similar between *Acomys* and *Mus*

We first set out to test if baseline differences in circulating peripheral white blood cell (WBC) profiles existed prior to injury in *Acomys* and *Mus*. Using a Sudan Black B modified Giemsa-Wright stain, we quantified monocytes, lymphocytes, neutrophils and eosinophils from *Acomys* and *Mus* whole blood (*Figure 1A–D*). Both species exhibited similar profiles and typical morphologies for all four cell types (*Figure 1A–E*). For instance, monocytes were distinguishable by their kidney-shaped nucleus and diffuse cytoplasmic stain (*Figure 1A*), while lymphocytes were similar in size to RBCs and their compact nucleus filled the entire cell (*Figure 1B*). Polymorphonuclear neutrophils stained strongly with Sudan-Black B and displayed multi-lobed nuclei (*Figure 1C*). In contrast, while eosinophils displayed multi-lobed nuclei and dark pink granules in the cytoplasm they contained few if any Sudan-Black-stained granules (*Figure 1D*). In *Mus* and *Acomys*, the percentage of circulating lymphocytes was significantly higher than other leukocyte populations, and eosinophils comprised the smallest population of circulating leukocytes (*Figure 1E*) (Tukey's Multiple comparison, simple effect p<0.05, *Figure 1—source data 1*). These data are consistent with other leukocyte profiles from outbred CD1 mice showing that lymphocytes comprise the highest percentage of circulating WBCs (*Hedrich, 2004*). Importantly, while we identified differences in the percentage of leukocyte subtypes within each species, leukocyte profiles were the same between *Acomys* and *Mus* (two-way ANOVA, species effect F = 0.01, p=0.92, and leukocyte subtype effect F = 97.04, p<0.0001, n = 8 *Acomys*; n = 4 *Mus*). Our data demonstrate these two rodent species possess the same circulating leukocyte profiles prior to injury and provide a baseline to ask if local differences arise following injury.

### The kinetics of inflammatory cell accumulation are generally similar between regeneration and scarring

Building on our observation that circulating leukocyte populations are similar in *Acomys* and *Mus*, we next assessed the acute inflammatory reaction to injury during epimorphic regeneration and scarring using our 4 mm punch assay through the ear pinna (*Gawriluk et al., 2016*; *Seifert et al., 2012a*) (*Figure 2—figure supplement 1A–C*). To quantify the influx of myeloid cells into the injured ear tissue, we performed fluorescence-activated cell sorting (FACS) using CD11b (*Figure 2A–C*). CD11b (aka ITGAM) is a broad-spectrum marker used to isolate mammalian macrophages and

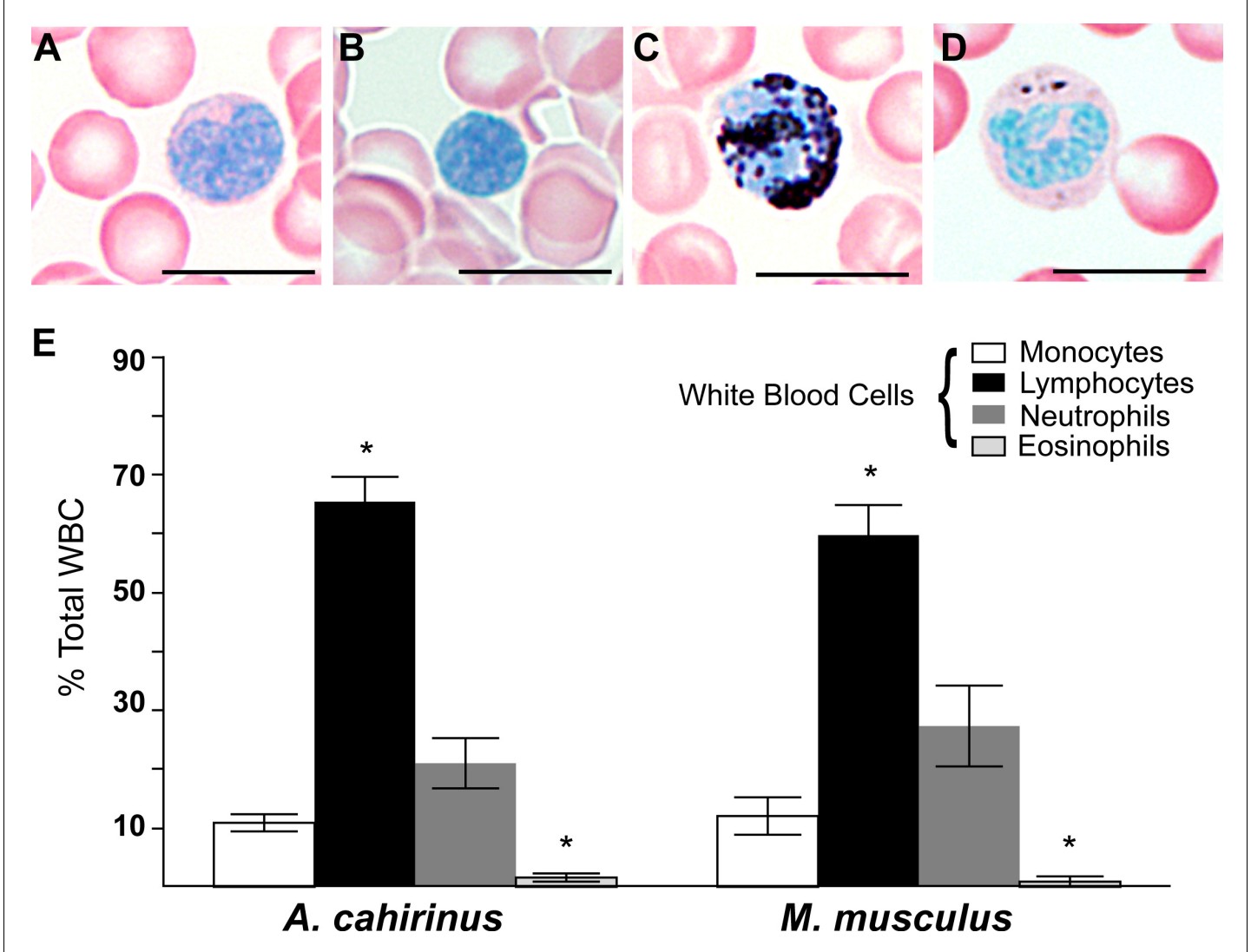

**Figure 1.** Circulating leukocyte profiles from uninjured animals are the same in *A. cahirinus* and *M. musculus*. (A–D) Sudan-Black B modified Giemsa Wright stain helps identify leukocyte subtypes based on morphology and stain in blood smears of *A. cahirinus*. Monocytes (A) show kidney shaped nucleus and diffuse cytoplasmic stain; lymphocytes (B) show round nuclei encompassing most of the cell with very little cytoplasm; polymorphonuclear neutrophils (C) show dense black staining of cytoplasmic granules and a banded, multi-lobed nucleus; and eosinophils (D) show dense pink staining of cytoplasmic granules and multi-lobed nucleus. Scale bar = 10 μm. (E) Counts of white blood cell subtypes as a percentage of total white blood cells (two-way ANOVA for main effects species F = 0.01, p=0.92; and leukocyte subtype effect F = 97.04, p<0.0001. *Tukey's multiple comparison test for simple effect leukocyte subtype p<0.05 indicating significant differences when comparing neutrophils versus lymphocytes, neutrophils versus eosinophils, lymphocytes versus monocytes, and lymphocytes versus eosinophils within each species; S.E.M.; n = 8 *Acomys*; n = 4 *Mus*).

The following source data is available for figure 1:

**Source data 1.** Statistical values are reported for comparing circulating leukocytes between and within species.

neutrophils across a range of species from mouse to humans (*Figueiredo et al., 2013*; *Sawano et al., 2001*; *Tamatani et al., 1991*; *Venneri et al., 2007*; *Venosa et al., 2015*). Using our recently published regeneration transcriptome for *Acomys*, we found *Cd11b* was upregulated after injury (*Gawriluk et al., 2016*). Alignment of *Acomys* and *Mus Cd11b* revealed 88% nucleotide identity compared to a 79% identity between *Mus* and Human (*Table 1*). FACS analysis using CD11b isolated a specific cell population in *Acomys* and *Mus* (*Figure 2A–B*). While we observed a significant increase in CD11b+ cells in response to injury in both species (two-way ANOVA with main effect

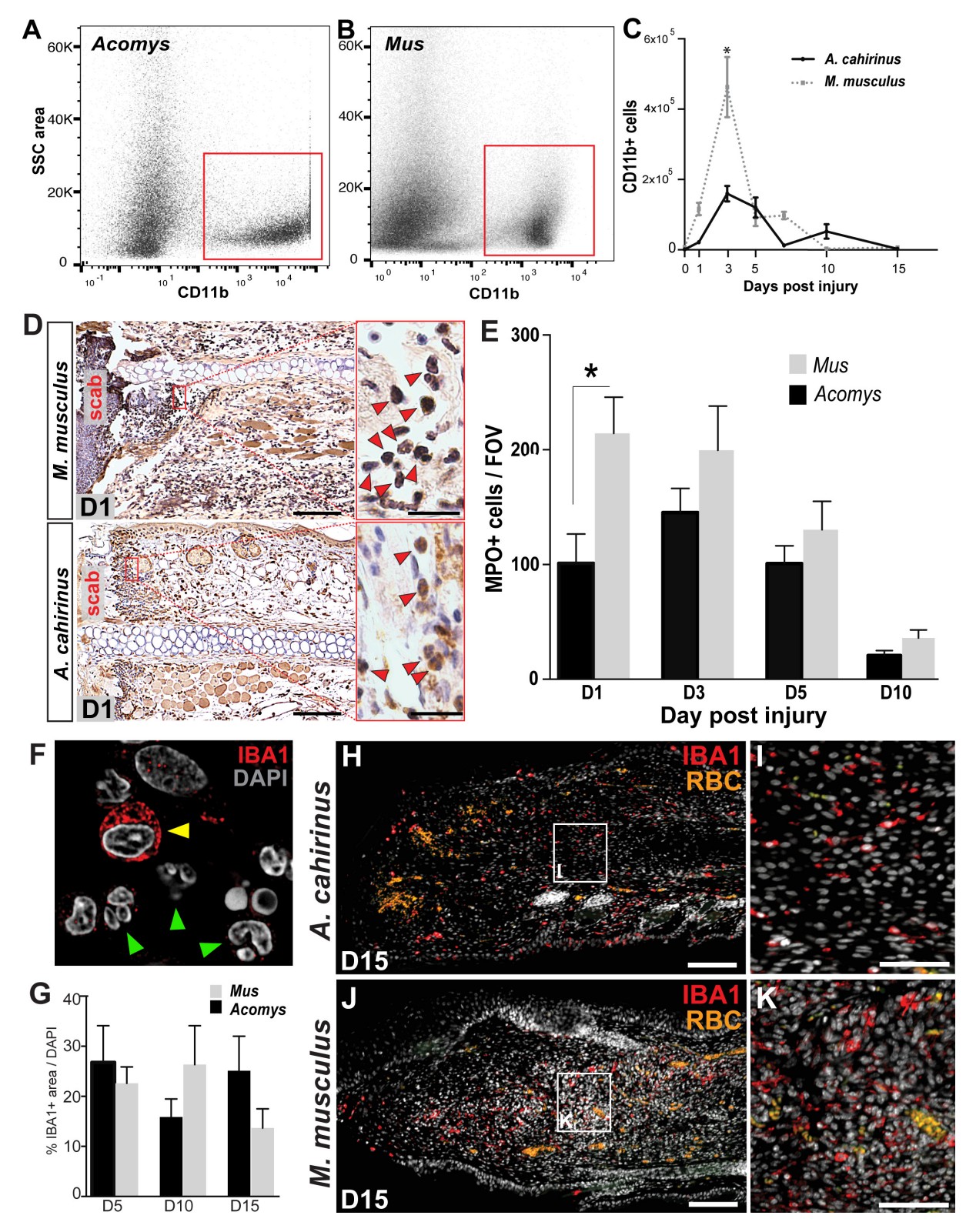

**Figure 2.** Acute infiltration of neutrophils and macrophages is a hallmark of regeneration and scarring. (A–B). Single-cell suspensions of whole tissue isolates from injured ears at D5 subjected to flow cytometry using CD11b show two distinct populations of cells, one CD11b- and one CD11b+ (red boxes) in *Acomys* (A) and *Mus* (B). (C) Quantifying cells over time using flow cytometry shows a peak increase of CD11b+ cells in *Mus* at D3 and a broader but smaller peak of CD11b+ cells in *Acomys* between D3 and 5 (two-way ANOVA main effect time F = 31.86, p<0.0001, main effect species, *Figure 2 continued on next page*

*Figure 2 continued*

F = 17.02, p=0.0002, *Sidak's multiple comparison test p<0.05, n = 4 animals combined left and right ear/species per timepoint). (**D**) Representative images of immunohistochemistry for myeloperoxidase (brown) in *M. musculus* (top panel) and *A. cahirinus* (bottom panel) 24 hr post injury. Nuclei (blue) were counterstained with Mayer's Hematoxylin. Magnification 200x, Scale bars = 50 µm. Inset images highlight polymorphonuclear appearance of positively stained cells (red arrows). Scale bars = 20 µm. (**E**) Cell counts of polymorphonuclear/MPO+ cells in healing tissue per field of view (FOV) (n = 5 animals/species, D1; n = 6 animals/species, D3, D5; n = 4 animals/species, D10, two-way ANOVA, main effect time and species F = 11.12, p<0.0001, F = 8.229, p=0.007 respectively, *p<0.05 Sidak's multiple comparisons test at time points indicated). (**F**) Myeloid protein IBA1 (red) reactivity in *Acomys* ear tissue at D5 showing distinct positive cells (yellow arrow) and negative cells with multi-lobed nuclei characteristic of neutrophils (green arrows). DAPI = grey, IBA1 = red. (**G**) Quantification of the total IBA1+ area in *Acomys* and *Mus* ears at D5, D10 and D20 normalized to total DAPI+ area (n = 3/species, two-way ANOVA, main effect time and species F = 0.132, p=0.723, F = 0.438, p=0.655, respectively) (**H–K**) Representative images of the IBA-1+ area quantified in (**G**) at D15. IBA-1+ cells localize proximal and distal to the injury site in *Acomys* and *Mus* and within the blastema in *Acomys*. IBA-1 = red, DAPI = grey, autofluorescent red blood cells (RBC) = orange. Scale bars (**H,J**) = 100 µm. Scale bar (**I,K**) = 50 µm Distal = left, Dorsal = top of image.

The following figure supplements are available for figure 2:

**Figure supplement 1.** Timeline of regeneration following 4 mm ear punch injury in *Acomys* compared to scar-formation in *Mus*.

**Figure supplement 2.** Isolation of monocytes using flow cytometry.

time F = 31.86, p<0.0001 and species F = 17.06, p=0.0002), the acute increase at D3 was significantly greater in *Mus* than *Acomys* (Sidak's multiple comparison test p<0.05) (*Figure 2C*).

Because CD11b isolates macrophages and neutrophils, we next sought to individually quantify neutrophil and macrophage influx at the injury site in *Acomys* and *Mus*. We first used the well-characterized cell surface Ly6G antigen to separate neutrophil and macrophage populations in *Mus* (*Bain et al., 2014*; *Mirza et al., 2009*; *Rose et al., 2012*) and detected a clear population of Ly6G+ cells (*Figure 2—figure supplement 2A*). This population of Ly6G+ cells was significantly elevated in healing tissue between D1-D7, with peak numbers occurring at D3 (*Figure 2—figure supplement 2B*). Whereas Ly6G+/CD11b+ cells dominated the injury site at D3 in *Mus* (*Figure 2—figure supplement 1C* -blue dots), these cells were replaced by a Ly6G-/CD11b+ population at D7 (*Figure 2—figure supplement 2C* - red dots). Thus, during the initial wave of leukocyte recruitment in *Mus* (D1-5), CD11b+ cells are primarily neutrophils and at later stages of inflammation (D7-D15) CD11b+ cells are primarily macrophages (*Figure 2—figure supplement 2D*). Interestingly, we did not detect a clear Ly6G+ population of CD11b+ cells from either tissue or circulating blood isolated from *Acomys* (*Figure 2—figure supplement 2A*).

The failure to resolve a discrete population of Ly6G+ cells in *Acomys* suggested neutrophils might more closely resemble rat and human neutrophils which lack Ly6G (*Lee et al., 2013*). The mouse *Ly6*

**Table 1.** Nucleotide comparisons for protein targets used in this study. Comparison is between *Mus* and *Acomys* and *Mus* and Human.

| *Mus* gene name | *Acomys* | Human |
| --- | --- | --- |
| *Ly6g* (lymphocyte antigen 6 complex, locus G) | No homolog | No homolog |
| *Ly6e* (lymphocyte antigen 6 complex, locus E) | 100% | 69% |
| *Cd11b* (integrin alpha M) | 88% | 79% |
| *Iba1/Aif1* (allograft inflammatory factor 1) | 86% | 85% |
| *Cd86* (CD86 antigen) | 78% | 76% |
| *Cd206* (mannose receptor, C type 1) | 90% | 81% |
| *Arg1* (arginase 1) | 81% | 78% |
| *Cd3e* (CD3 antigen, epsilon polypeptide) | 80% | 74% |
| *Mpo* (myeloperoxidase) | 94% | 85% |
| *Adgre1* (adhesion G protein-coupled receptor E1), (F4/80) | 86% | 79% |

gene complex is composed of 11 known *Ly6* genes. Of these, the subcluster *Ly6b*, *c*, *g* and *e* are most highly expressed by mouse neutrophils, whereas rat and human lack all these genes except *Ly6e* (*Lee et al., 2013*). While we identified the genomic sequence and expressed transcript for the *Acomys* homolog of *Ly6e* (*Table 1*), examination of the *Ly6* gene complex using our preliminary *A. cahirinus* genome revealed no homologs for *Ly6b*, *Ly6c* or *Ly6g* (*Table 1*). These data reveal that the genomic structure for the Ly6 complex in *Acomys* is similar to rat and human suggesting that *Acomys* neutrophils more closely resemble rat and human neutrophils which also lack *Ly6g* (*Lee et al., 2013*).

Given these results, we turned to the neutrophilic marker MPO which reliably detects polymorphonuclear cells in tissue sections across species (*Bradley et al., 1982*; *Petrie et al., 2014*; *Seifert et al., 2012b*). We assessed MPO reactivity at D1, D3, D5 and D10 to compare neutrophil accumulation and clearance in *Mus* and *Acomys* (*Figure 2D,E*). Neutrophils were readily identified as MPO+ with multi-lobed nuclei (*Figure 2D*), and *Mus* displayed a significantly higher number of neutrophils 24 hr after injury when compared to *Acomys* (two-way ANOVA with main effects, species F = 8.229, p=0.007 and time F = 11.12, p<0.0001; Sidak's multiple comparison test for simple effect between species, p<0.05 at D1) (*Figure 2E*). After D1, *Mus* and *Acomys* showed comparable numbers of neutrophils at the site of injury with a return to baseline levels by D10 (*Figure 2E*, Sidak's multiple comparison test for simple effect between species, p>0.05 at D3-10). In both species, neutrophils initially accumulated distal to the cut in the dermis and periodically exhibited signs of cell death (e.g. pyknotic nuclei and reduction in cell size). We also observed neutrophils within the scab at all timepoints analyzed (*Figure 2D*). Taken together, our results show that both species exhibit neutrophil infiltration, accumulation and clearance in response to injury, although neutrophils accumulate faster, at higher levels and appear to be more active 24 hr after injury in *Mus.*

Concomitant with neutrophil invasion, circulating and tissue-specific macrophages are activated in response to injury and are recruited to the injury site at the outset of regenerative and scarring responses (*Godwin et al., 2013*; *Li et al., 2012*; *Nguyen-Chi et al., 2015*; reviewed in *Novak and Koh, 2013*; *Petrie et al., 2014*; *Varga et al., 2016*). To ascertain total macrophage abundance in healing tissue, we employed the pan-macrophage marker, ionized calcium-binding adaptor molecule 1 (IBA-1), to quantify the spatiotemporal pattern of macrophage infiltration in *Acomys* and *Mus*. IBA-1 is an actin-binding protein active in macrophages and microglia and has been used to label cells in mouse, dog, cat, human and other primates (*Imai et al., 1996*; *Köhler, 2007*; *Pierezan et al., 2014*; *Sasaki et al., 2001*; *Schmidt et al., 2016*). In *Acomys*, we found IBA-1+ cells in the mesenchyme and the epidermis, but IBA-1 was specifically absent from polymorphonuclear neutrophils (*Figure 2F*, arrows). We found no significant difference in the amount of IBA-1+ cells between *Mus* and *Acomys* at D5, D10 or D15 (*Figure 2G*, two-way ANOVA main effect species F = 0.132 p=0.723, and main effect time F = 0.438 p=0.655). Accumulation of IBA1+ cells occurred at similar levels in both species within 200 µM proximal to and distal to the site of injury (*Figure 2H–K*). Notably, IBA-1+ cells were present within the blastemal region of D15 ears in *Acomys* (*Figure 2H,I*) as well as within the central granulation tissue region of D15 ears in *Mus* (*Figure 2J,K*). These data demonstrate that macrophages persist at the injury site during regeneration and scarring up to 2 weeks after injury.

## Acute inflammation during regeneration is characterized by high ROS production

To determine the extent of the inflammatory reaction in vivo, we measured myeloperoxidase (MPO) activity and reactive oxygen species (ROS) production (*Tseng and Kung, 2012*). To track MPO activity in vivo we used luminol, a chemiluminescent compound that exhibits specific and high sensitivity for phagocyte-mediated MPO activity from neutrophils (*Gross et al., 2009*). Due to its larger size and reduced cell permeability (compared to luminol), we used lucigenin to measure ROS production by NADPH oxidase. Lucigenin chemiluminescence is primarily attributed to macrophage activation, and to a lesser extent from endothelial cell and neutrophil populations (*Tseng and Kung, 2012*). Tracking luminol chemiluminescence, *Mus* displayed peak MPO activity between 24–48 hr after injury (*Figure 3A*, repeated measures ANOVA F = 5.095, p<0.001). This finding was congruent with our immunohistochemical cell count data using MPO (*Figure 2E*). While *Acomys* displayed a similar peak in MPO activity, the magnitude was significantly muted compared to *Mus* (*Figure 3A*, *Sidak's multiple comparison test p<0.05 for time points indicated). Conversely, *Acomys* exhibited a much

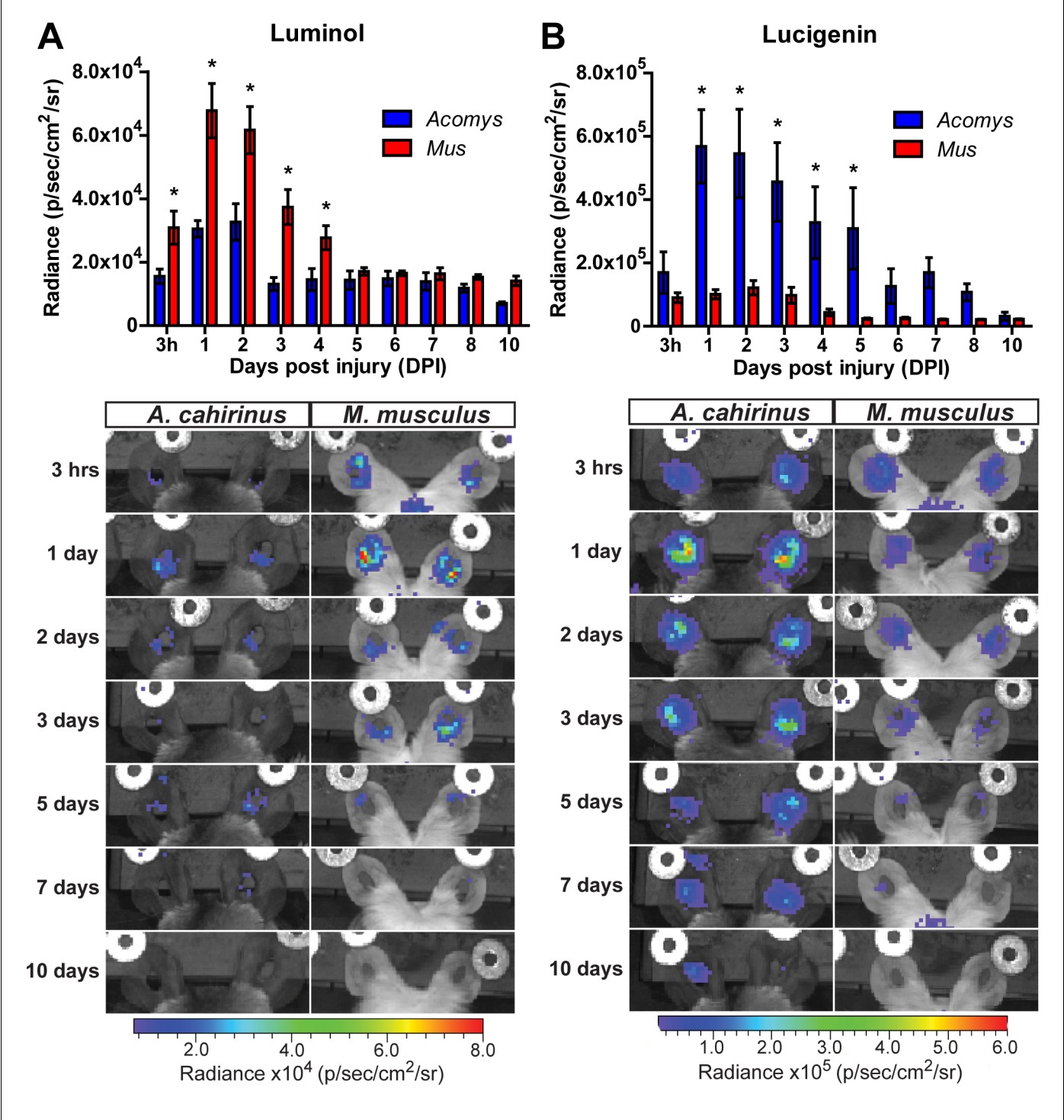

**Figure 3.** Acute myeloperoxidase activity is elevated during scarring, while reactive oxygen species production is elevated during regeneration. (A–B) In vivo imaging of the chemiluminescent compounds luminol and lucigenin, showing myeloperoxidase activity (A) or ROS production (B) in the injured ears of *Mus* (red bars) and *Acomys* (blue bars). Images below graphs are representative for each timepoint. Chemiluminescence is measured in radiance [photons (p) per second (s) emitted from a square centimeter of tissue ($cm^2$) and radiating into a solid angle of one steradian (sr)]. For luminol experiments: n = 7 *Mus* (14 ears) and n = 6 *Acomys* (12 ears) repeated measures ANOVA, F = 5.095, p<0.001 and for lucigenin experiments: n = 8 *Mus* (16 ears) and n = 6 *Acomys*, (12 ears), repeated measures ANOVA F = 4.536, p<0.001, *p<0.05 Sidak's multiple comparison test between species at the time indicated.

more robust production of ROS compared to *Mus* during the entire inflammatory phase (*Figure 3B*, repeated measures ANOVA F = 4.536, p<0.001). Lucigenin chemiluminescence peaked 24 hr after injury in *Acomys* and remained significantly elevated compared to *Mus* through D5 (*Figure 3B*, *Sidak's multiple comparison test p≤0.05 for the time indicated). Throughout the inflammatory phase, lucigenin chemiluminescence was significantly muted in *Mus* (*Figure 3B*). Together, these data suggest a bias toward strong neutrophilic MPO activity during scarring and early and prolonged macrophage-produced ROS during regeneration.

## Macrophage depletion inhibits blastema formation and regeneration

Persistence of macrophages at the injury site, coupled with stark differences in ROS production suggested that macrophages might positively affect regeneration consistent with observations during *Xenopus* tail regeneration (*Love et al., 2013*). In vivo depletion of macrophages using clodronate liposomes during axolotl limb regeneration and via genetic ablation in zebrafish caudal fins supports a requirement for these cells to stimulate a regenerative response (*Godwin et al., 2013*; *Petrie et al., 2014*). Clodronate liposomes have been widely used to deplete systemic and local populations of phagocytes, the majority of which are macrophages (reviewed in *van Rooijen and Hendrikx, 2010*). In order to test if macrophages also regulate epimorphic regeneration in spiny mice, we depleted these cells by injecting clodronate liposomes (Clo-Lipo) at the base of the ear immediately prior to injury (D0), at D2, and at D5 (*Figure 4A*). Control animals received similar injections of PBS liposomes (control) (*Figure 4A*). Whereas control ear holes initiated a regenerative response and began closing at D5, Clo-Lipo injected animals did not initiate ear hole closure until D20 (*Figure 4B–C*). In contrast to control ears, which showed complete ear hole closure by D34, all Clo-Lipo injected ears remained open past D44 (*Figure 4B,C*). Complete hole closure and regeneration among Clo-Lipo-treated ears was delayed, with 3/12 closed by D53, 8/12 ears closed by D60 and 11/12 ears closed by D70 (*Figure 4B–C*). Although blastema formation and expansion occurred in PBS-Lipo-treated ears, examination of Clo-Lipo-treated ears showed a sharp reduction in cell accumulation past the original cut site anda delay in blastema formation at D20 (*Figure 4D–E*). The early stages of blastema formation were evident in 50% of macrophage-depleted ears (4/8) beginning at D20 (*Figure 4D–E*, dotted line denotes original plane of biopsy).

In Clo-Lipo-treated ears, we noted a slight expansion of ear hole area and residual scabbing at D10 suggesting defects in re-epithelialization following macrophage depletion (*Figure 4B–C*). Indeed, examining macrophage-depleted ear pinna at D5 we observed a lack of cartilage histolysis (*Figure 4F*, green arrow) and a delay in re-epithelialization (*Figure 4F*, blue arrows). This delay in re-epithelialization was coincident with an accumulation of neutrophils (*Figure 4G* yellow arrowheads) when compared to control ears (*Figure 4H–I*). Monocytic cells were apparent throughout the tissue in control ears (*Figure 4H* green arrowheads) but were absent in Clo-Lipo treated ears (*Figure 4F*). We confirmed effective depletion of macrophages from the injury site by staining for the pan macrophage marker IBA-1 (*Figure 4J–L*). Macrophage populations can be restored within 2 weeks of the final Clo-Lipo injection (*Ames et al., 2016*; *Li et al., 2013*; *Summan et al., 2006*; *Sunderkötter et al., 2004*), and IBA-1+ cells at D20 reveals a return of macrophages in Clo-Lipo-treated ears, concurrent with re-epithelialization and initiation of blastema formation (*Figure 4M–N*). Together, these data support the important activity of macrophages to facilitate histolysis and re-epithelialization during the early phase of regeneration. Furthermore, our data suggests that macrophages directly or indirectly are necessary for blastema formation during regeneration in spiny mice.

## Activated CD86+ macrophages are restricted from the regeneration blastema

Macrophages maintain an interesting duality as professional phagocytes and as coordinators of the local immune response. Because macrophages were present and persistent during regeneration and scarring, it was possible that macrophage phenotype might contribute to the different healing outcomes. First, we asked whether spiny mice macrophages could undergo classical (M1) and alternative (M2) activation under stereotypical conditions (*Figure 5*). We isolated spiny mouse bone marrow as previously described for *Mus* (*Edwards et al., 2006*) and activated these cells using Macrophage-Colony Stimulating Factor (M-CSF) to produce activated bone-marrow-derived macrophages (BMDM) (*Figure 5*). After 1 week in culture, all BMDM were CD11b+ (*Figure 5A*).

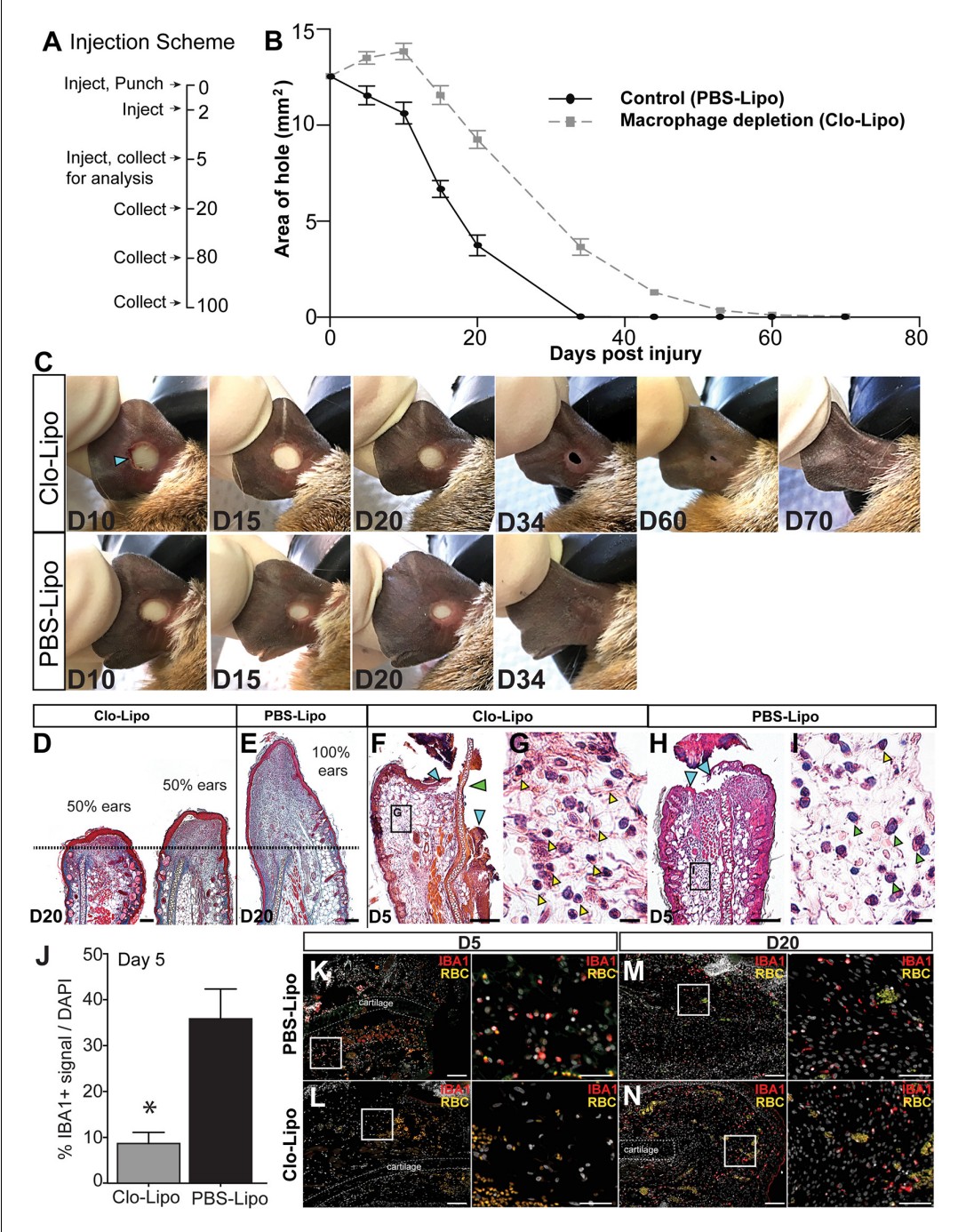

**Figure 4.** Macrophage depletion with clodronate liposomes inhibits regeneration. (**A**) Ears were injected with clodronate liposomes (Clo-Lipo) or PBS liposome controls (PBS-Lipo) at D0 immediately before injury, D2 after injury and at D5. Ears were allowed to regenerate and tissue collected at later time points. (**B**) Wound size was measured over time. PBS-lipo ears close completely by D34 (black line, graph, bottom panel images). Clo-Lipo ears remain open until D70 (grey dotted line, graph, top panel images, n = 6 animals, 12 ears per treatment). (**C**) Representative ear from Clo-Lipo (top panel) and PBS-Lipo (bottom panel) followed over time. (**D–E**) H & E stained Clo-Lipo ear at D20 shows variable reduction in cell accumulation past the injury site (black dotted line) with relatively little new growth compared to PBS-Lipo ears. Scale bar = 100 μm. (**F–I**) H & E stained Clo-Lipo ear at D5 (**F**) shows a delay in epidermal closure (blue arrows) and loss cartilage plate histolysis (green arrow) compared to PBS-Lipo ears (**H**). Scale bar = 200 μm. (**G**) Boxed region in (**F**) showing accumulation of polymorphonuclear cells (yellow arrowheads). (**I**) Boxed region in (**H**) with monocytic cells evident (green arrowheads) and few polymorphonuclear cells present (yellow arrowhead). Scale bar = 10 μm. (**J–N**) IBA-1 immuno-positive area following macrophage depletion compared to control. Total IBA-1+ area normalized to total DAPI+ area at D5 after the final treatment (**J**, n = 3 animals per treatment, *unpaired Student's t-test p<0.05). IBA-1+ cells in PBS-lipo tissue at D5 (**K**) compared to Clo-lipo tissue at D5 (**L**). IBA1+ cells in PBS-Lipo

*Figure 4 continued on next page*

*Figure 4 continued*

tissue at D20 (**M**) and Clo-lipo treated tissue at D20 (**N**) show a return of positive cells.. Scale bar = 100 µm. Box delineates area of high-magnification images, Scale bar = 50 µm. IBA-1 = red, DAPI = grey, autofluorescent red blood cells (RBC) = orange. Distal = left, Dorsal = top.

Next, we tested if *Acomys* macrophages could undergo classical and alternative activation (polarization) toward an M1 or M2 phenotype, respectively. Classic activation assays stimulate BMDMs with the pro-inflammatory molecules interferon gamma (IFNγ) and lipopolysaccharide (LPS) which specifically increase the expression of CD86 among other pro-inflammatory cell surface markers and cytokines (*Edwards et al., 2006*; *Hathcock et al., 1994*; *Inaba et al., 1994*). In response to IFNγ and LPS, *Acomys* BMDMs remained CD11b+ and upregulated CD86 compared to un-stimulated and M2 stimulated macrophages (*Figure 5B–F*). Alternative activation with interleukin 4 (IL4) increases the expression of CD206 (*Stein et al., 1992*) and Arginase 1 among other pro-reparative, cytoprotective genes. In response to IL-4, *Acomys* BMDMs remained CD11b+ and expressed CD206 (*Figure 5C,I*). In contrast, most unstimulated and classically activated BMDMs were CD206- (*Figure 5G–H*). In addition to CD206, stimulation with IL-4 also elicited Arginase 1 reactivity in BMDMs (*Figure 5F*). We also isolated and activated BMDMs from *Mus* and confirmed they display a similar pattern of protein regulation in response to polarization (*Figure 5J–R*). In addition to these phenotypic markers, we also tested the mouse pan macrophage marker F4/80 (*Austyn and Gordon, 1981*) for its reactivity to *Acomys* macrophages (*Figure 5—figure supplement 1*). Using unstimulated BMDMs, we only found a small subset of F4/80+ cells in *Acomys* compared to ubiquitous labeling in *Mus* (*Figure 5—figure supplement 1A,D*). Following IFNγ+LPS stimulation, we observed an increase in F4/80+ *Acomys* macrophages, whereas IL-4 stimulation did not increase F4/80 labeling compared to unstimulated BMDMs (*Figure 5—figure supplement 1B–C*). Examining F4/80 staining in vivo, we found that F4/80+ cells were distinct from CD206+ cells in Acomys (*Figure 5—figure supplement 1G*). In contrast, we found that F4/80+ cells were mostly CD206+ in *Mus* supporting its localization as a pan macrophage marker in *Mus* (*Figure 5—figure supplement 1H*). Thus, our in vitro and in vivo findings suggest that F4/80 only marks a subset of macrophages in *Acomys* (most likely M1). Despite this difference in F4/80 reactivity, our in vitro results show intrinsic similarities between *Acomys* and *Mus* macrophages supporting a general capacity to activate and change phenotype given specific wound contexts and stimuli.

Next, we addressed whether differences in the spatiotemporal distribution of macrophage subtypes might promote regeneration in lieu of scarring. First, we quantified accumulation of classically activated macrophages in healing tissue by staining for CD86 (*Figure 5E* and *Figure 6A–B*). Using immunohistochemistry to localize CD86+ cells in *Acomys* and *Mus*, we determined these cells were rare in uninjured ear tissue (*Figure 6—figure supplement 1A–F*). Although rare, in *Mus*, we did find CD86+-positive cells in the epidermal layer reminiscent of human Langerhan's cells (*Boltjes and van Wijk, 2014*) and in the perichondrium (*Figure 6—figure supplement 1A–F*), whereas in uninjured *Acomys* tissue CD86+ cells were restricted to the dermis (*Figure 6—figure supplement 1D–E*). These cells had long, thin projections and a spindle-like shape characteristic of dermal dendritic cells found in humans (*Boltjes and van Wijk, 2014*).

We next assessed the distribution of CD86+ cells during regeneration and scarring. Comparing CD86+ cells distal to the injury between species, we found a significant effect of time (two-way ANOVA F = 131.8, p<0.0001) and species (two-way ANOVA F = 220.0, p<0.0001) whereby injury elicited a strong increase in CD86+ cells per unit area in *Mus*, but not in *Acomys* at D3 (Sidak's multiple comparison test p<0.05) (*Figure 6A*). Examining the spatial distribution of CD86+ cells in *Mus*, we found that they accumulated in the connective tissue distal to the cut cartilage (*Figure 6B*). Co-staining with the T-cell surface marker CD3 revealed interactions between CD86+ and CD3+ cells in this region as well as in the epidermis, consistent with the role of CD86 as a co-stimulatory molecule for T-cell activation (*Figure 6B*). In contrast to *Mus*, CD86+ macrophages behaved very differently in *Acomys* during regeneration. Following injury, we did not observe CD86+ cells accumulating in the blastema (*Figure 6B* and *Figure 6—figure supplement 1J*). Instead, at D15 we observed clusters of CD86+ cells located almost exclusively lateral to the cut cartilage (*Figure 6B*). Thus, while CD86+ cells are present in *Acomys*, they remain proximally restricted and do not infiltrate the blastema,

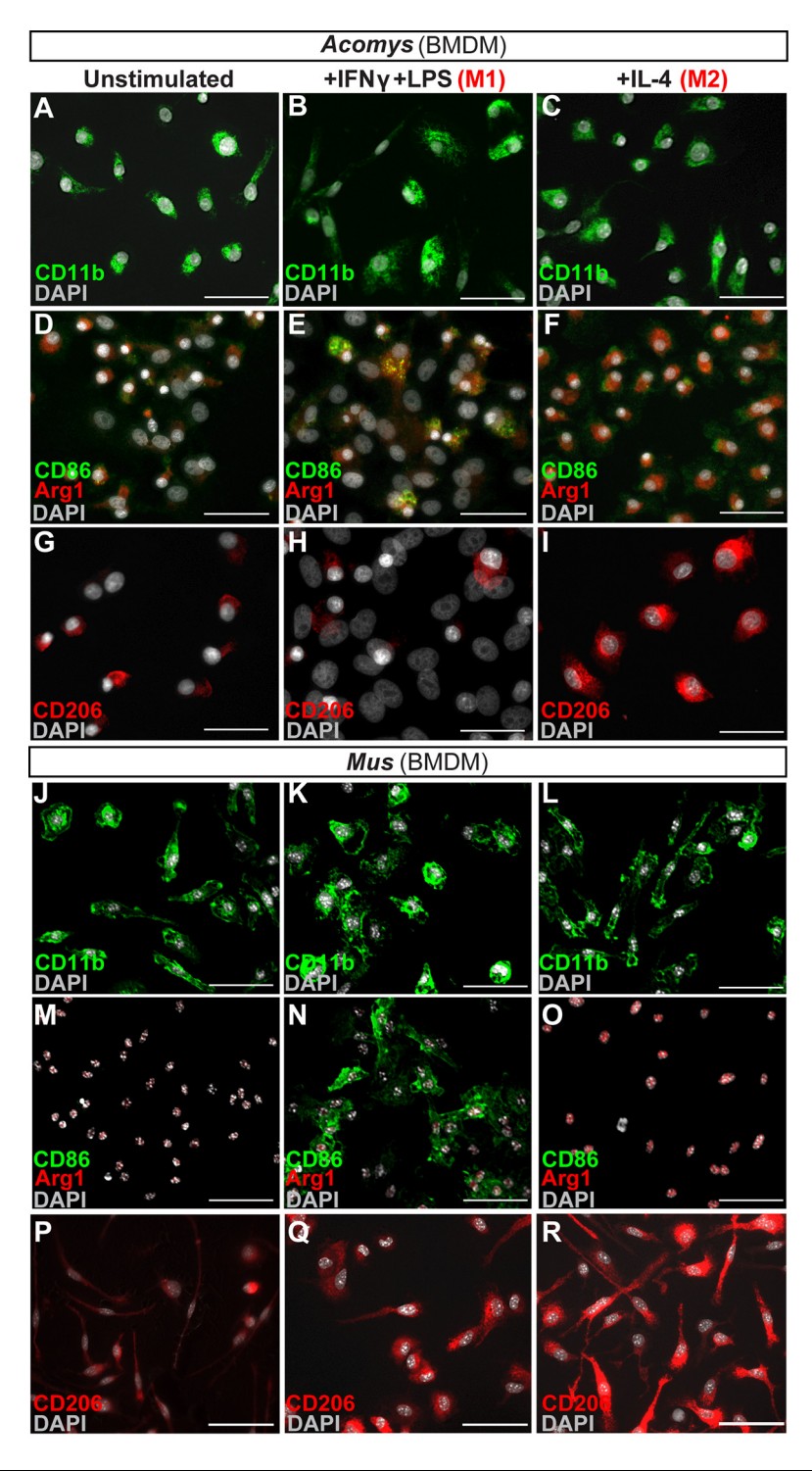

**Figure 5.** In vitro activation assays shows *Acomys* macrophages can be polarized to express different markers. (A–I) Bone-marrow-derived macrophages isolated from *Acomys* femurs are cultured with no cytokines (unstimulated, A, D, G) with IFNγ+LPS (M1, B, E, H) or with IL-4 (M2, C, F, I). Immunocytochemistry for the pan-macrophage marker CD11b (green) (**A–C**), for the M1 macrophage marker CD86 (green) and the M2 macrophage marker Arginase 1 (red) (**D–F**), or CD206 (red) (**G–I**). (**J–R**). Bone-marrow-derived macrophages were isolated from *Mus* femurs and cultured with no cytokines (J, M, P) with IFNγ and LPS (K, N, Q) or with IL4 (L, O, R) as above. Immunocytochemistry was performed for CD11b (green) (**J–K**), for CD86 (green) and Arginase 1 (red) (**M–O**), and

*Figure 5 continued on next page*

*Figure 5 continued*

CD206 (red) (**P–R**). Nuclei were counterstained with DAPI (grey) in all panels. Scale bars = 50 μm. Images are representative of n = 3 technical replicates.

The following figure supplement is available for figure 5:

**Figure supplement 1.** Immunofluorescent staining for macrophage marker F4/80 in *Acomys* and *Mus*.

whereas these same cells in *Mus* accumulate in newly deposited connective tissue (granulation tissue).

Although alternatively activated (M2) macrophages are usually associated with pro-regenerative outcomes, this concept remains largely untested for epimorphic regeneration (*Campbell et al., 2013*; *Kigerl et al., 2009*; *Wang et al., 2014*). Using our comparative system, we tested for an association between CD206+ (M2) macrophages and a regenerative response. Prior to injury, CD206+ cells were spatially distributed throughout the ear in similar numbers in both species (*Figure 6—figure supplement 2A–D*). These cells were present in the dermis between the epidermis and elastic cartilage of the ear (*Figure 6—figure supplement 2A–D*). We did not detect CD206+ cells in the epidermis in either species. Quantifying CD206+ cells distal to the injury in *Mus* and *Acomys*, we observed no significant difference across time (two-way ANOVA, F = 2.330, p=0.125) or between species (two-way ANOVA, F = 1.540, p=0.23) (*Figure 6C*). After D3, CD206+ cell numbers remained constant in *Acomys* and *Mus* and showed no significant change distal to the injury site (*Figure 6C*). Unlike CD86+ cells that localize proximal to the injury in *Acomys*, CD206+ cells were observed distal to the injury site in *Acomys* (and *Mus*) (*Figure 6D* and *Figure 6—figure supplement 2E–H*). However, the distal distribution of these cells in *Acomys* appeared regionalized with a CD206+ dense region directly beneath the epidermis and a CD206+ sparse region in the central blastema region of the injury (*Figure 6D*). In *Mus*, CD206+ cells were evenly distributed throughout the connective tissue distal to the injury (*Figure 6D*). Together with our data for IBA-1, our results show that while macrophages infiltrate the injury area during regeneration, the blastema is relatively free of classically activated macrophages.

## Discussion

A popular hypothesis to explain why most mammals heal injuries with scar tissue is that they evolved a strong inflammatory and adaptive immune response that induces intense fibrosis in lieu of regeneration (*Godwin, 2014*; *Mescher et al., 2017*). Yet, the fact that some mammals exhibit epimorphic regeneration (e.g. rodent and primate digit tips, rabbit and spiny mice ear punches and skin) (*Borgens, 1982*; *Gawriluk et al., 2016*; *Goss and Grimes, 1975*; *Han et al., 2008*; *Joseph and Dyson, 1966*; *Neufeld and Zhao, 1993*; *Seifert et al., 2012a*; *Singer et al., 1987*) suggests that regeneration can occur despite a complex adaptive immune system. Different immune system components and inherent physiological differences between mammals and traditional regeneration models like salamanders, newts and zebrafish (e.g. homeothermy versus poikilothermy, high versus low metabolic rates, etc.) make it difficult to extrapolate how inflammation and immunity might act to affect regeneration in mammals. There have been few studies detailing how the immune system responds during epimorphic regeneration and thus how it compares to immune mediated fibrosis. In this study, we have begun to address how inflammatory cells behave during complex tissue regeneration in African spiny mice (*Acomys*). We identified key differences in the spatiotemporal infiltration of inflammatory cells in regenerating and scar-forming systems and demonstrated that reactive oxygen species (ROS) catalyzed through NADPH oxidation were significantly increased at the injury site during regeneration. Importantly, we showed that macrophages were required for regeneration to proceed. These results support a role for inflammatory cell signals polarizing the injury response toward two very different outcomes.

Our characterization of circulating leukocyte profiles in *Acomys* and *Mus*, demonstrate that inherent differences in myeloid cell numbers do not explain intrinsic differences in regenerative ability. This contradicts a recent report suggesting that spiny mouse blood is neutropenic and that lower numbers of circulating neutrophils are responsible for a muted inflammatory response to injury

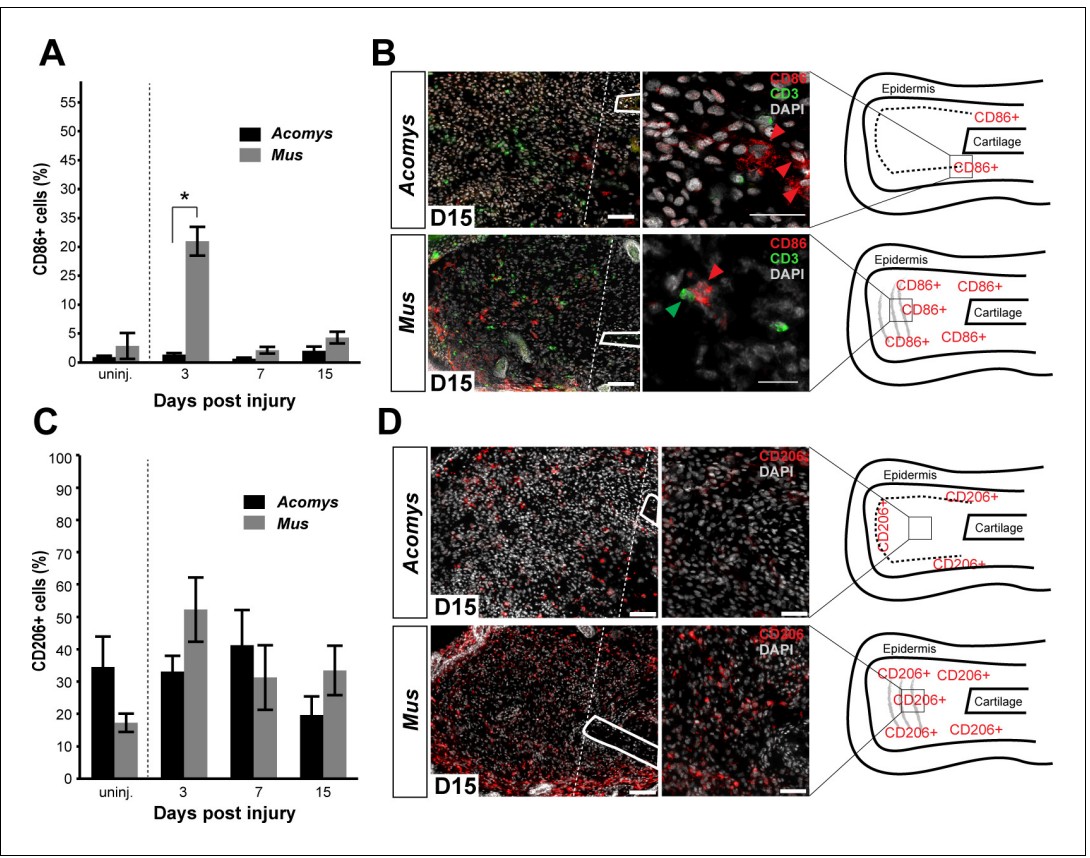

**Figure 6.** Activated CD86+ macrophages are restricted from the blastema in *Acomys*. (**A**) Immunofluorescent staining for the cell surface marker CD86 in *Acomys* and *Mus* at specific time points after injury as a percent of total area analyzed (n = 4 per time point; two-way ANOVA main effect time and species F = 131.8, p<0.0001, F = 220.0, p<0.0001 *Sidak's multiple comparison test p<0.05 at time point indicated). (**B**) Left panels 10x magnification, CD86 (red), DAPI (grey). Scale bar = 100 μm. White lines delineate cartilage. Middle panel, 40x magnification, CD86 (red), DAPI (grey). Scale bar = 20 μm. Right panel, diagram with the general overview of CD86+ localization at D15 and delineating high-magnification area (box). (**C**) Immunofluorescent staining for the cell surface marker CD206 in *Acomys* and *Mus* as a percent of total area analyzed (n = 4 per time point per species, two-way ANOVA main effect time and species, F = 2.33, p=0.125, F = 1.54, p=0.230 respectively). (**D**) Left panel magnification 10x. CD206 (red), DAPI (grey). Scale bar = 100 μm. White lines delineate cartilage. Middle panel 20x magnification. CD206 (red), DAPI (grey). Scale bar = 50 μm. Right panel, Schematic depicting the general trends for CD206+ cell localization in *Acomys* and *Mus* at D15. Box delineates the location of the high-magnification images.

The following figure supplements are available for figure 6:

**Figure supplement 1.** Immunofluorescent staining for CD86+ cells at D0, D3 and D7 post injury in *Mus* and *Acomys*.

**Figure supplement 2.** Immunofluorescent staining for CD206+ cells at D0, D3, D7 post injury in *Mus* and *Acomys*.

(*Brant et al., 2016*). In fact, our immunohistochemical data comparing neutrophil infiltration in *Acomys* and *Mus* show that while neutrophils accumulate faster during scarring, peak neutrophil numbers are equivalent 3 days after injury in both species. In general agreement with this data, we observed differences in the intensity of myeloperoxidase activity via luminol chemilumenscence persisting until D4 after which time there was no difference across species. While neutrophil invasion in response to injury is apparent across all regenerating vertebrates (*Jordan and Speidel, 1924*; *Li et al., 2012*; *Seifert et al., 2012b*), our results suggest that precise activity levels could lead to

different injury outcomes. While prolonged inflammation can antagonize regeneration (*Margalit et al., 2005*; *Mescher et al., 2013*), our data supports an initial wave of cell-based inflammation as a shared feature of any injury response, including regeneration.

Along with neutrophils, monocytes also infiltrate local tissue after injury, and converging evidence suggests that macrophages are required in some capacity for epimorphic regeneration (*Godwin et al., 2013*; *Petrie et al., 2014*). Our results extend these vertebrate studies by demonstrating a similar requirement during epimorphic regeneration in mammals. Acute depletion of phagocytic monocytes with clodronate liposomes delayed the initiation of ear hole closure and blastema formation by up to 2 weeks. Importantly, re-commencement of blastema formation was concurrent with the return of macrophages into the injured tissue. Examination of ear tissue at D5 revealed that phagocyte depletion delayed re-epithelialization and histolysis, two key events that are themselves required for regeneration. It is possible that macrophages provide an initiating signal for regeneration or remove subpopulations of local cells secreting inhibitory signals (e.g. senescent cells). In support of the first idea, ROS production has been suggested as an essential early signal for regeneration based on studies in *Xenopus* and zebrafish tail models of regeneration (*Gauron et al., 2013*; *Love et al., 2013*). Macrophages are a major source of ROS after injury, and we observed significantly stronger and prolonged ROS production during regeneration compared to scarring (*Weber et al., 2016*). Understanding the functional consequences of balanced ROS production through NADPH oxidation versus myeloperoxidase activity will require a more complete understanding of what cell types are responsible for ROS production and how ROS more specifically can affect local cellular phenotypes.

In support of the idea that macrophages may limit inhibitory signals through selective removal of senescent cells, recent work in salamanders suggested that clearance of senescent cells is important for limb regeneration (*Yun et al., 2015*) and persistence of senescent cells during liver regeneration leads to excessive fibrosis (*Krizhanovsky et al., 2008*). Furthermore, the accumulation of senescent cells with age has been suggested to shorten lifespan, degrade tissue function, and increase the expression of pro-inflammatory cytokines in mammals (*Baker et al., 2016*, *2011*). These and other studies suggest that proper clearance of senescent cells from damaged tissues may promote regenerative outcomes. Interestingly, as the induction of cellular senescence occurs during normal wound healing when a scar forms, it is possible that clearance of senescent cells is less important than the secretory phenotype of these cells which in some contexts can promote regeneration. For instance, short exposure to factors secreted by senescent cells in response to injury decreases fibrosis and promotes stem cell gene expression (*Chiche et al., 2017*; *Jun and Lau, 2010*; *Ritschka et al., 2017*). These studies underscore the importance of analyzing how senescent cells regulate regeneration and scarring and provide evidence that the phenotype of senescent cells and their timely removal by macrophages could be an important factor in *Acomys* ear regeneration.

While our data demonstrates that macrophages as a total population are required for regeneration in mammals, a requirement for macrophage subtypes and how these cells interact with other immune cells during epimorphic regeneration is not known. Because the activation of macrophages is generally associated with collagen production and fibrotic disease (*Duffield et al., 2005*; *Gibbons et al., 2011*; *Wynn, 2008*), the question remains as to how these immune cells orchestrate both regeneration and scar-formation in different species. Previous studies suggest a temporal component to macrophage function as a major factor for determining the outcome to skin, liver and bone injury (*Alexander et al., 2011*; *Arnold et al., 2007*; *Duffield et al., 2005*; *Mirza et al., 2009*). While we did not observe temporal differences in total macrophage accumulation between *Acomys* and *Mus* in the ear pinna, we did observe distinct differences in the spatial distribution of macrophage subtypes. For instance, while M2 (CD206+) macrophages were present at comparable levels in *Acomys* and *Mus*, we observed a zone dense in CD206+ macrophages in *Acomys* that were associated with de novo hair follicle development and a zone sparse in CD206+ cells that were associated with an area of decreased collagen production and blastema formation. Whereas CD206+ cells were absent from the blastema in *Acomys*, in *Mus*, we observed CD206+ cells throughout the collagen-rich granulation tissue, a situation similar to fibrosis observed during skin wound repair (*Mirza and Koh, 2011*; *Song et al., 2000*; *Willenborg et al., 2012*). The spatial restriction of CD206+ cells suggests unique interactions between macrophages and surrounding cells may drive differences in a pro-regenerative or pro-fibrotic environment.

On the other hand, it is possible that the local environment (cells and/or ECM) in regenerating systems drives immune cell profile instead of immune cells driving injury outcomes. We noted an accumulation of M1 macrophages (CD86+ cells) in injured *Mus* tissue, and the ability of CD86+ cells to interact with CD3+ T-cells is consistent with the role of CD86 as a co-stimulatory molecule that promotes T-cell activation and production of pro-inflammatory cytokines (*Hathcock et al., 1994*; *Lanier et al., 1995*; *Peng et al., 2013*). Conversely, the *Acomys* blastema was mostly devoid of M1 macrophages and we did not observe CD86+/CD3+ interacting cells. This was not due to an inability to detect M1 macrophages in *Acomys*, as we readily observed these cells at the boundary between injured and uninjured tissue. Moreover, we found that *Acomys* macrophages up-regulated CD86 in vitro when classically activated in response to IFNγ and LPS stimulation (*Ding et al., 1993*; *Freedman et al., 1991*). This shows that *Acomys* macrophages possess the intrinsic ability to express M1 markers (e.g. CD86) in response to pro-inflammatory stimuli, but may be inhibited from doing so within the blastema in vivo.

A recent report suggested that spiny mice regenerate skin because they lack a robust inflammatory response (*Brant et al., 2016*). While our findings contradict their primary conclusions, careful re-interpretation of their data supports the idea that inflammation occurs during regeneration and scarring, and that different inflammatory cell phenotypes polarize the response to injury. Brant et al. based their conclusions on three primary findings. First, they presented circulating leukocyte profiles showing spiny mice with lower neutrophils and higher lymphocytes in circulation when compared to CD1 mice and suggested this would lead to fewer neutrophils in spiny mice wounds. As we have shown, neutrophils indeed arrive at the wound bed and numbers are not significantly different three days post injury. Interestingly, our data for myeloperoxidase activity suggests that differences in neutrophil activity, rather than simple numerical differences, likely play a more important role during the injury response. Second, they were unable to detect macrophages in spiny mice wounds using F4/80 and concluded no macrophages infiltrated regenerating skin wounds. Using a cross-species marker for total macrophages (IBA1), we show that macrophages indeed infiltrate an injury site. More importantly, we demonstrate that macrophages are *required* for regeneration. Instead, differential infiltration of macrophage subtypes exists between regeneration and scarring. While F4/80 may be a pan-macrophage marker in mice, studies have shown that this is not the case for humans (*Hamann et al., 2007*), and it remains unclear if this is the case for other mammals. Our data shows that F4/80 marks a subset of macrophages with pro-inflammatory characteristics in *Acomys* (i.e. F4/80+ cells are increased with M1 stimulation in vitro and F4/80+ cells do not colocalize with the M2 marker, CD206, in vivo). When re-interpreted with this knowledge, their results align with our findings that reduced numbers of pro-inflammatory macrophages enter the wound site during regeneration or are present, but not activated toward a pro-inflammatory phenotype. Thirdly, using a cytokine array designed against mouse antigens they detected 30 cytokines during acute inflammation (D0-D7 post injury) in *Mus*, but only detected 12 of these cytokines in spiny mice during the same period. They interpreted the absence of 18 cytokines as an absence of these signals in spiny mice wounds. However, without validation of epitopes, failure to detect particular spiny mouse cytokines on a *Mus*-specific cytokine array does not reflect an absence of spiny mouse antigen. Moreover, of the 12 spiny mouse cytokines they did detect, 5 are classic pro-inflammatory markers (e.g. IL-1α, IL-1ß, IL-1ra, CXCL13 and IFNγ) that were upregulated in response to injury and present at similar levels to *Mus*. Thus, their spiny mouse cytokine data demonstrates strong acute inflammation in response to injury. Although nuances between regenerating dorsal skin wounds and complex tissue of the ear pinna are likely to exist, fine-tuning inflammatory responses is key to promoting scar-free outcomes.

Of the mammalian models of epimorphic regeneration that exist, very few have investigated how inflammatory cells affect blastema formation and regeneration. A forthcoming study on digit tip regeneration lends support to our conclusion that macrophages are required to help initiate regeneration (*Muneoka et al., 2017*). Autoimmune-prone MRL mouse strains and their parent strain LgJ have been promoted as a mammalian regeneration model (*Clark et al., 1998*). Paradoxically, while these strains possess enhanced rates of healing, published reports across most injury models have demonstrated that they do not regenerate (*Colwell et al., 2006*; *Gawriluk et al., 2016*; *Moseley et al., 2011*; *Smiley et al., 2014*). Studies have documented that the healing response of the MRL mouse differs depending upon the type of injury, be it the ear pinna or other tissues (*Beare et al., 2006*; *Colwell et al., 2006*; *Davis et al., 2007*; *Kench et al., 1999*; *Rajnoch et al.,*

*2003*; *Tolba et al., 2010*). In the ear specifically, although 2 mm biopsy punches close and form new cartilage nodules (*Clark et al., 1998*), larger holes and a 2 mm crush injury heal with excessive collagen deposition producing a scar (*Gawriluk et al., 2016*; *Rajnoch et al., 2003*). In our view, the use of MRL strains as purported regeneration models has obscured their potential utility for studying inflammation and fibrosis. Of interest, MRL mice have been extensively studied for their aberrant macrophage profiles (*Dang-Vu et al., 1987*; *Donnelly et al., 1990*; *Santoro et al., 1988*). Bone marrow and peritoneal macrophages proliferate without CSF stimulation (*Hamilton et al., 1998*), a rare observation among other strains of mice, and accumulation of macrophages is commonly observed in MRL tissues (*Bloom et al., 1993*; *Davis and Lennon, 2005*; *Yui et al., 1991*). Macrophages in strains of MRL mice show higher production of $H_2O_2$ suggesting higher pro-inflammatory activity (*Dang-Vu et al., 1987*) and show increased expression of *Tgfβ1*, a growth factor implicated during increased fibrosis (reviewed in *Border and Noble, 1994*; *Kench et al., 1999*). Thus, MRL macrophages and their response to injury are unique in many respects. A detailed study of the immune response in MRL mice could instruct how inflammation guides fibrotic repair based on intrinsic and environmental tissue differences.

Macrophage phenotypes change based on environmental cues (*Stout et al., 2005*) and macrophages that infiltrate injured tissue exhibit temporal changes in gene and protein expression (*Arnold et al., 2007*; *Varga et al., 2016*). In addition, cancer and mesenchymal stem cells (MSCs) can drive macrophage phenotypes dampening the production of pro-inflammatory cytokines (reviewed in *Gao et al., 2016*). These observations underscore the fluid nature of macrophage phenotypes in response to injury and disease and support a model where different subtypes differentially affect healing outcomes. Although further work is needed to clarify macrophage activation phenotypes during regeneration, our results support the initial inflammatory environment as a potential source of pro-regenerative signals. Future studies in spiny mice will need to determine how specific immune cell types signal to local cells, whether macrophages induce change in the ECM and if these cells can be manipulated in a non-regenerative system. Future studies into the role of specific macrophage and other immune cell phenotypes will resolve the role of these cells in scar formation and regeneration.

## Materials and methods

### Animal care, 4 mm ear punch and tissue collection

*Acomys cahirinus* and *Mus musculus* (Swiss Webster Envigro_Harlan Hsd:ND4) were housed at the University of Kentucky, Lexington, KY. *A. cahirinus* were housed at a density of 10–15 individuals in metal wire cages (24 in. x 18 in. x 16 in., height x width x depth) (Quality Cage Company, Portland, OR) and fed a 3:1 mixture by volume of 14% protein mouse chow (Teklad Global 2014, Harlan Laboratories, Indianapolis, IN) and black-oil sunflower seeds (Pennington Seed Inc., Madison, GA) 1x/day (*Haughton et al., 2016*). *Mus* were fed mouse chow only. *Acomys* and *Mus* mice were exposed to natural light, and all animals used were sexually mature. Experiments used a combination of male and female animals matched between species. For ear punch, animals were anesthetized with 3% vaporized isoflurane (v/v) (Henry Schein Animal Health, Dublin, OH) at 1 psi oxygen flow rate. A 4 mm biopsy punch (Sklar Instruments, West Chester, PA) was used to create a through-and-through hole in the center of the right and left ear pinna. Ear tissue was collected at specified time points with an 8 mm biopsy punch (Sklar Instruments, West Chester, PA) circumscribing the original injury. All animal procedures were approved by the University of Kentucky Institutional Animal Care and Use Committee (IACUC) under protocol 2013–1119.

### Flow cytometry

Healing ear tissue was harvested as outline above at D0, 1, 3, 5, 7, 10 and 15. To create a single-cell suspension, both ears were combined into one tube for each animal, and we used enzymatic and mechanical digestion as previously described with modifications (*Jensen et al., 2010*). Briefly, tissue was digested with a 1:1 trypsin, dispase solution for 1 hr at 37°C allowing for subsequent mechanical separation of the epidermis. Separated epidermis and dermis were incubated with a solution of collagenase (1 mg/mL, VWR RLMB120-0100), hyaluronidase (0.5 mg/mL, VWR IC1512780), and elastase (0.015 U/μL, VWR IB1753-MC) in HBSS (VWR 45000–456) for 1 hr at 37°C. Following digestion, cell

suspensions were washed with PBS and filtered through a 70 µm cell strainer. Single-cell suspensions were incubated with an FcγR block (CD16/32 block, 20 µg/mL, BD Pharmingen Cat# 553141) followed by incubation with directly conjugated primary antibodies at 4°C for 1 hr. Antibodies included APC-conjugated Ly6G (BD Pharmingen Cat# 580599, 3 µg/mL), PE-conjugated CD11b (BD Pharmingen, Cat# 557397, 3 µg/mL), diluted in FBS-staining buffer (BD Pharmingen, Cat# 554656). Fluorescent-activated cell sorting (FACS) was carried out by trained experts in the University of Kentucky Flow Cytometry Core using the iCyt Synergy sorter system (Sony Biotechnology Inc., San Jose, CA). Laser calibration and compensation was performed for each experiment using unstained, single fluorescent, and fluorescent minus one (FMO) control samples. Dot plots were created using FloJo (version 10) (n = 4 animals per timepoint).

## Analysis of circulating leukocytes

Blood was collected from the submandibular venous bed of age and gender matched *Acomys* (n = 8) and *Mus* (n = 4). Individuals were anesthetized with 4% (v/v) isoflurane and gently scruffed so that the skin covering the submandibular venous bed was taut. A 5 mm lancet (Medipont Inc., Mineola, NY) was inserted quickly into the venous bed and one drop of blood was collected per slide, with a total of 3 drops per animal. Slides were allowed to air dry and were prepared for a Sudan Black B modified Giemsa-Wright staining as described by (*Sheehan and Storey, 1947*) using formaldehyde vapor fixation for 10 min. Slides were incubated for 1 hr in a filtered Sudan Black B (3 mg/mL, Sigma Aldrich, St. Louis, MO) disodium hydrogen phosphate solution. Slides were incubated with Giemsa-Wright (Sigma Aldrich, St. Louis, MO) stain followed by a 1 min wash in 0.5% acetic acid. Slides were allowed to air dry and coverslipped with CytoSeal XYL. Images were collected on an Olympus BX51 upright microscope at 40x magnification. Ten fields of view were acquired per slide for an average 9224 total cells per animal and an average 42 total white blood cells per animal. White blood cells were hand counted based on granular staining and nuclear morphology. Cell subtype is reported as a percent of total WBC per animal (n = 4 *Mus*, n = 8 *Acomys*).

## Intra-vital ROS measurements

On the days indicated, animals with 4 mm wounds were anesthetized using 2.5% (v/v) isoflurane and injected I.P. with lucigenin (5 mg/kg in PBS; M8010 Sigma-Aldrich, St. Louis, MO) or luminol (100 mg/kg in PBS; A4685 Sigma-Aldrich). After approximately 10 min of incubation, the animals were imaged with a chemiluminescent, in vivo imaging-system (IVIS 200 Spectrum; Perkin Elmer, Waltham, MA). We determined that the peak activity occurs within 20 min post injection, and thus, measured luminescence during the first 25 min post injection. We acquired 15 images with a 60 s exposure, f-stop equal to 1, binning factor equal to 8, and a 21.6 cm field of view. To analyze the radiance emitted from the wounds, a circular region of interest with a diameter of 6 mm was placed around each ear wound and the total flux in the regions was measured. The maximum value for each wound over the 15 images was used for subsequent analyses.

## Histology and immunohistochemistry

Harvested tissue was placed into 10% (v/v) neutral buffered formalin (American Master Tech Scientific Inc., Lodi, CA) and incubated at 4°C overnight. Tissue was washed three times with PBS, three times with 70% (v/v) ethanol and stored at 4°C in 70% (v/v) ethanol. All tissue processings were completed using a rapid microwave histoprocessor (Micron Instruments, Inc. Carlsbad, CA). Tissues were embedded in paraffin (Leica Biosystems, Buffalo Grove, IL) and 5-µm sections were placed onto Superfrost Plus slides (Fisher Scientific). Immunohistochemical and H & E staining were performed on deparaffinized and rehydrated sections. Immunohistochemical staining for rabbit anti-human MPO (Dako, Cat #A0398) was carried out as previously described (*Gawriluk et al., 2016*) using heat-mediated antigen retrieval in a Tris-EDTA buffer at pH9.0. For, secondary detection of the primary antibody, we used biotin-conjugated goat anti-rabbit antibody followed by HRP-conjugated streptavidin for 3,3'-diaminobenzidine conversion according to Vector Elite ABC staining kit (Vector, Burlingame, CA). Nuclei were counterstained with Mayer's hematoxylin for brightfield visualization and coverslips mounted with Cytoseal XYL (ThermoFischer, Waltham, MA).

Immunofluorescent staining was performed on flash frozen tissue collected in Tissue-Tek OCT (Sakura, Torrance, CA) and frozen on dry ice. 8 µm sections were collected on a Leica CM1900

cryostat at 20°C. Tissue was subsequently fixed to slides with ice-cold acetone for 15 min and washed in PBS. Slides were incubated with primary antibodies overnight: rat anti-mouse CD86 (1:100, BD Biosciences, Cat #553698), goat anti-mouse CD206 (1:1000, R and D Systems, Cat #AF2535), rabbit anti-human CD3 (1:400, Dako, Cat #A0452), rabbit anti-mouse IBA1 (Wako, Cat #019–19741). Secondary detection of antibodies was carried out using donkey antibodies conjugated to Alexa Fluor 488, 594 (1:500, Invitrogen, Carlsbad, CA) Nuclei were counterstained with 10 µg/ml DAPI and coverslips were mounted using ProLong Gold mounting medium (Invitrogen, Carlsbad, CA) for fluorescence.

To quantify the total area of positive signal for fluorescent images, three photomicrographs within the center of the injury were obtained at 40x magnification using an Olympus BX53 fluorescent deconvolution microscope (Olympus America Inc). Quantification of positive signal was performed on four separate samples (one ear per animal) per time point (unless otherwise noted) by thresholding fluorescent signal and mask subsampling with Metamorph Imaging software (Molecular Devices, Sunnyvale, CA). The ratio of total immuno-positive area per total area of the region of interest was then calculated. To quantify total number of DAB-positive cells, two photomicrographs within the center of the injury were obtained at 20x magnification using an Olympus BX53 microscope. Counts were calculated using the Cell Counter plugin for ImageJ (version 1.51d). Cells included in counts were based on both nuclear morphology and immuno-positive staining. Total positive cells were reported as a percent of total area (pixels) in a region of interest that excluded scab, epidermis, and cartilage plate. The ID of the samples being run was blinded from the user until the end of the analysis.

## In vitro macrophage analysis

Macrophage progenitors were isolated from femur and tibia of *Acomys* and *Mus*. After sacrifice, femur and tibia were surgically removed, and mechanically cleared of all skin, muscle and tendon. Marrow was aspirated from bones by flushing the marrow with 10 mL of RPMI +10% FBS through a 28 gauge syringe. Red blood cells were lysed with a hypotonic solution and remaining cells were plated at a density of $1 \times 10^6$ cells/mL in T-75 culture flasks. For the first 7 days, bone marrow cells were grown in complete RPMI media supplemented with 20% L929 media containing M-CSF, 10% FBS, 1% PenStrep. After 7 days in culture, BMDM were split using cold PBS and a cell scraper, plated onto coverslips in 24-well plates at a density of $5 \times 10^5$ cells/mL and allowed to settle for 24 hr in complete RPMI media supplemented with 10% FBS and 1% PenStrep. For macrophage activation, cells were stimulated with either 500 µL of IFNγ (20 pg/mL) and LPS (200 ng/mL) in cRPMI media 10% FBS or 500 uL of IL-4 (20 ng/mL) in cRPMI media 10% FBS. 24 hr after activation, media was removed, cells washed once with cold PBS and fixed with 10% neutral buffered formalin for 10 min at room temperature. After three washes with PBS to remove excess formalin, cells were permeabilized with 1% Triton x100 in PBS for 10 min. Immunocytochemistry was performed by first blocking non-specific binding sites using 5% goat serum in PBS block. Cells were washed once with PBS and incubated with primary antibody overnight at 4°C. Primary antibodies include rabbit anti mouse Arginase 1 (1:500 GeneTex, Cat #113131), goat anti mouse CD206 (1:1000, R and D Systems, Cat #AF2535), rat anti-mouse CD86 (1:100, BD Biosciences, Cat #553689), rat anti-mouse Cd11b (1:500, Abd Serotec, Cat #MCA74G), rabbit anti-mouse IBA1 (1:1000, Wako, Cat #019–19741), rat anti-mouse F4/80 (1:400, clone BM8, eBiosciences, Cat #14-4801-82). Detection of primary antibodies was accomplished by incubating cells with secondary antibody conjugated to Alexa Fluor fluorophores (1:1000, Invitrogen, Carlsbad, CA) as follows: Donkey anti-Goat AF546, Goat anti-Chicken AF488, Donkey anti-Rat 488, and Donkey anti-Rabbit 546. Cell nuclei were counterstained with DAPI and after air-drying, coverslips were mounted to slides using ImmunoMount (Invitrogen, Carlsbad, CA).

## Clodronate-liposome injections

Following previously reported protocols (*Barrera et al., 2000*; *Li et al., 2013*; *van Rooijen and Hendrikx, 2010*), we inject 20 µL of 50 µg/mL clodronate liposomes- or PBS liposomes (www.clodronate-liposomes.org, (*van Rooijen and Hendrikx, 2010*)) as vehicle control. Injections were performed using a Hamilton syringe (Hamilton Company, Reno Nevada) at the base of each ear at D0, 2 and 5 after injury. Tissue was collected at D5 (n = 3 *Acomys*/treatment) and D20 (n = 4

*Acomys*/treatment) for immunohistochemical and histological examination. Ear hole closure was measured over time using a digital micrometer (n = 6 *Acomys*, 12 ears per treatment).

## Statistics

For analyzing immuno-positive cells, observations within the same mouse were averaged so analysis could be on the level of the experimental unit. For each marker, we analyzed the percentage of immune-positive signal as a fraction of total DAPI positive area using a two-way ANOVA with species, day and the species*day interaction). When noted, Sidak's or Tukey's multiple comparison tests were conducted where appropriate using Prism 6 Data analysis software (Graphpad Software Inc, La Jolla, CA). Sample size was determined by calculating power from previous immunohistochemical studies. These calculations show with appropriate transformation of percentages and a standard deviation of 0.07, groups of n = 4 animals should be sufficient to detect a species effect with a size of 0.16395 with 80% power and alpha = 0.05. To calculate sample size for repeated measures, we used previous wounding studies to determine maximum standard deviation of 2.21. These tests show groups of n = 5 animals, 10 ears should be sufficient to detect a treatment effect of 2.77 with 80% power and alpha = 0.05. To analyze the chemiluminescent data, we performed a repeated-measure, two-way ANOVA with time and species as main effects and Sidak's post-hoc tests to compare the species x time effect. Graphs were created using Prism and annotated in Illustrator (Adobe Creative Suite 6, San Jose, CA). Graphs display standard error of mean (S.E.M.) with statistical significance values indicated in figure legends. Sample size (n) is stated in each figure legend and refers to biological replication size (n = number of distinct animals) with the exception of in vitro studies in which n = number of technical replicates for BMDM activation.

## Acknowledgements

The authors would like to thank Jennifer Strange and Greg Bauman of the University of Kentucky Flow Cytometry and Cell Sorting Core Facility, which is supported in part by the Office of the Vice President for Research, the Markey Cancer Center and an NCI Center Core Support Grant (P30 CA177558) to the University of Kentucky Markey Cancer Center. We thank Jeremiah Smith, Shishir Biswas, and Chanung Wang for help with *A. cahirinus* transcriptomic and genomic data, and members of the Seifert and Gensel labs for insightful discussion. This work was supported by a grant from the National Science Foundation (NSF) and the Office for International Science and Engineering (OISE) (IOS −1353713) to AWS. JS is supported by a University of Kentucky Postdoctoral Research Fellowship.

## Additional information

### Funding

| Funder | Grant reference number | Author |
|---|---|---|
| National Science Foundation | Office for International Science and Engineering IOS 1353713 | Ashley W Seifert |
| University of Kentucky | Postdoctoral Fellowship | Jennifer Simkin |

The funders had no role in study design, data collection and interpretation, or the decision to submit the work for publication.

### Author contributions

JS, Conceptualization, Data curation, Formal analysis, Validation, Writing—original draft, Writing—review and editing; TRG, Conceptualization, Data curation, Formal analysis, Writing—review and editing; JCG, Funding acquisition, Writing—review and editing; AWS, Conceptualization, Supervision, Funding acquisition, Validation, Writing—original draft, Writing—review and editing

### Author ORCIDs

Jennifer Simkin, http://orcid.org/0000-0002-6122-4222
Thomas R Gawriluk, http://orcid.org/0000-0001-9868-9704

John C Gensel, http://orcid.org/0000-0001-8980-108X
Ashley W Seifert, http://orcid.org/0000-0001-6576-3664

## Ethics

Animal experimentation: All animal procedures were performed with approval by the University of Kentucky Institutional Animal Care and Use Committee (IACUC) under protocol 2013-1119. Study design was in accordance with the Guide for the Care and Use of Laboratory Animals of the National Institutes of Health.

## Additional files

### Major datasets

The following previously published dataset was used:

| Author(s) | Year | Dataset title | Dataset URL | Database, license, and accessibility information |
|---|---|---|---|---|
| Gawriluk TR, Simkin J, Thompson KL, Biswas SK, Clare-Salzler Z, Kimani JM, Kiama SG, Smith JJ, Ezenwa VO, Seifert AW | 2016 | Comparing regeneration and fibrosis using Acomys and Mus | https://www.ncbi.nlm.nih.gov/geo/query/acc.cgi?acc=GSE71761 | Publicly available at the NCBI Gene Expression OmniBus (accession no: GSE71761) |

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
