## [Decision Letter]

Thank you for submitting your article "Macrophages are necessary for epimorphic regeneration in African spiny mice" for consideration by *eLife*. Your article has been favorably evaluated by Fiona Watt (Senior Editor) and three reviewers, one of whom, Alejandro Sánchez Alvarado (Reviewer #1), is a member of our Board of Reviewing Editors. The following individuals involved in review of your submission have agreed to reveal their identity: Luisa DiPietro (Reviewer #2); David Stocum (Reviewer #3).

The reviewers have discussed the reviews with one another and the Reviewing Editor has drafted this decision to help you prepare a revised submission.

Summary:

This is an interesting paper opening new areas of mammalian regeneration research using the spiny mouse Acomys. Mammals repair tissue by scarring rather than regeneration and it is currently believed that is likely due to a robust adaptive immune system that favors fibrosis over regeneration. This is supported by indirect evidence from non-mammalian vertebrates such as urodeles, which regenerate successfully and possess relatively undeveloped adaptive immune systems when compared to mammals. In this paper, the authors used ear punch assays in both *Acomys* and *M. musculus* to compare and contrast the macrophage responses in these animals during wound healing and regeneration. By directly comparing inflammatory cell activation in a 4 mm ear injury during regeneration (*Acomys cahirinus*) and scarring (*Mus musculus*), the authors uncovered that the profiles of circulating leucocytes are similar in each. Both species mount an acute inflammatory response that differed in that *Mus* had higher myeloperoxidase activity and *Acomys* had higher ROS activity, which could be related to scarring vs. regeneration. Curiously, the authors also noted that the spatial distribution of activated macrophage subtypes was unique during regeneration with pro-inflammatory macrophages failing to infiltrate the regeneration blastema.

The analysis of leucocyte populations is meticulous and the data are consistent with the authors' conclusions. This paper chops away some of the undergrowth surrounding the idea that a highly developed adaptive immune system determines that scarring predominates over regeneration. Clearly, there is no large-scale difference between *Mus* and *Acomys* immune cells that determines the difference. To the contrary, an immune response is characteristic of both fibrosis and regeneration. Other, factors, therefore, appear to be in play in determining whether the outcome to wounding in mammals results in regeneration or fibrosis.

Essential revisions:

1) Data presentation would improve if the authors were to provide a time-course schematic of the ear punch regeneration assay such that it could be used to follow the data presented. Although this sequence has been published, a summary diagram would be of great help in allowing the reader to interpret the data. This would let the reader easily understand the context of the time points examined in the current manuscript without having to dig up the prior paper.

2) The authors should spend considerably more time comparing, contrasting and explaining the differences that exist between the work reported in this manuscript ant the work of Brant et al. Additionally, a more scholarly discussion of the implications of the reported results and those of published findings for the MRL mouse should be attempted.

3) The data on macrophage polarization seems incomplete – what is the reason that the arginase assessment was not performed on *Mus*?

4) Neutrophils and macrophages have many functions in wound repair. They have bactericidal effects; macrophages are phagocytic; they release growth factors such as PDGF, TGF-β and HGF that promote epithelialization and fibroblast proliferation. Can the authors correlate the spatial findings here on macrophage location with growth factor production (or lack thereof), as assessed in previous papers?

5) The work of Maximinia Yun should be mentioned here. She has shown that macrophages in regenerating urodele limbs remove senescent cells and that this is obligatory for regeneration. The accumulation of senescent cells has also been implicated in the aging of organs; clearance of these cells preserves function (Baker et al., 2016). Removal of senescent cells could be a factor in *Acomys* ear tissue regeneration.

---

## [Author Response]

Essential revisions:

1) Data presentation would improve if the authors were to provide a time-course schematic of the ear punch regeneration assay such that it could be used to follow the data presented. Although this sequence has been published, a summary diagram would be of great help in allowing the reader to interpret the data. This would let the reader easily understand the context of the time points examined in the current manuscript without having to dig up the prior paper.

As per the reviewers' suggestion, we now include a time-course schematic as Figure 2—figure supplement 1. This new figure provides a graph (A) showing the mean kinetics of ear hole closure as a function of days post injury during regeneration in *Acomys* and scarring in *Mus*. The second panel (B) presents the process of regeneration following a 4 mm ear punch. It shows whole mount photographs of the ear pinna in both species during healing and aligns key processes (e.g., inflammation, re-epithelialization, blastema formation, etc.) as a function of days post injury. We believe this figure will assist the reader in following our experimental timelines and tie specific days to the overall injury response.

2) The authors should spend considerably more time comparing, contrasting and explaining the differences that exist between the work reported in this manuscript ant the work of Brant et al.

We appreciate being encouraged to expand our Discussion to include this comparison in our manuscript. We agree that it is important to explain the differences reported in our study compared to the Brant et al., (2016) study that examined scar-free healing of dorsal skin wounds in spiny mice. Brant et al., (2016) concluded based on their data that spiny mice lack a robust inflammatory response and that this could partly explain their ability to regenerate skin. This conclusion was based on their data showing comparatively low numbers of neutrophils in circulation, an inability to detect macrophages in the wound bed following injury and a failure to detect many inflammatory cytokines during wound healing. Briefly, these authors did not present neutrophil data in skin wounds and relied solely on their circulating leukocyte profiles to *infer* that neutrophil numbers would be reduced at the injury site. While we cannot explain the discrepancy in leukocyte counts, in our study we increased biological replicates for *Acomys* (n=8) and used a modified Wright-Giemsa stain to clearly differentiate neutrophils based on Sudan Black incorporation into granules. The increased resolution of Sudan Black increases confidence in our own counts. Their failure to detect macrophages is based on the assumption that F4/80 is a pan-macrophage marker in spiny mice. This assumption is incorrect, a conclusion we arrived at during our own study. We believe this information is important enough to include and we have now supplemented our manuscript with additional data in Figure 5—figure supplement 1. Our new data presents in vitro reactivity and in vivo co-localization data for F4/80 comparing *Acomys* and *Mus*. As is clear from our data, F4/80 is not a pan-macrophage marker in spiny mice, and its use as such by Brant et al., led to their conclusion that macrophages did not infiltrate skin wounds. In fact, it appears that F4/80 marks a subset of M1 macrophages in spiny mice. In this light, their data supports our finding that M1 macrophages as marked with CD86 or F4/80 do not infiltrate the blastema. Lastly, their spiny mouse cytokine data is based on a failure to detect over half of the cytokines included on the arrays they used. Instead of validating the arrays for spiny mice protein binding, they interpreted a lack of signal as an absence of the molecule. Again, a careful re-interpretation of some of their data actually lends support to our own conclusions. As per reviewers' request, we have distilled our comparisons and explanations of the differences between the two reports in a paragraph within the Discussion.

Additionally, a more scholarly discussion of the implications of the reported results and those of published findings for the MRL mouse should be attempted.

We appreciate the reviewers’ request for a more scholarly discussion of the MRL mouse and the published results purporting regeneration in this inbred strain. As the reviewers are no doubt aware, past and present work with this strain as it relates to regeneration has been fraught with controversy. In most instances, rigorous investigation of different injury paradigms have shown that, rather than regenerate, these mice exhibit increased inflammation and fibrosis in response to healing (e.g., skin – Colwell et al., 2006, Wound Repair and Regeneration;heart – Moseley et al., 2011, Journal of Pharmacy and Pharmacology, Smiley et al., 2014, Cardiovascular Pathology, etc.). However, there are reports supporting regeneration. Recently, our group demonstrated that MRL mice fail to close 4mm ear holes and instead produce fibrotic tissue to fill 2mm ear holes (Gawriluk et al., 2016. Nature Communications) a feat observed in other inbred strains (Reines et al., 2009, Rejuvenation Research, Kench et al., 1999, Clinical Immunology) and detailed for 2mm punch injuries in MRL mice (Rajnoch et al., 2003, Wound Repair and Regeneration). However, we believe that the MRL strain can be useful for understanding how inflammation directs fibrosis. Compared to normal mice, MRL mice exhibit: hematopoietic cells that overexpress TGFß1 at injury sites, increased macrophage recruitment to tissues during homeostasis and injury, and macrophages that produce higher levels of H_2_O_2_, etc. These and other inflammatory cell anomalies in this strain may drive the increased fibrosis reported following injury. Thus, in response to the reviewer’s request, we have amended our Discussion and added a paragraph that addresses published data for the MRL mouse related to our findings. We have also included discussion of how this model might be useful for understanding how inflammation can regulate fibrosis.

3) The data on macrophage polarization seems incomplete – what is the reason that the arginase assessment was not performed on Mus?

We had performed this analysis previously, but excluded the data for space considerations. In light of the reviewer’s request we have now included data for in vitro localization of Arginase 1 and CD11b in *Mus* for comparison to the parallel studies run in *Acomys*. This data is now included in revised Figure 5.

4) Neutrophils and macrophages have many functions in wound repair. They have bactericidal effects; macrophages are phagocytic; they release growth factors such as PDGF, TGF-β and HGF that promote epithelialization and fibroblast proliferation. Can the authors correlate the spatial findings here on macrophage location with growth factor production (or lack thereof), as assessed in previous papers?

We appreciate this comment and believe that quantifying growth factor production during regeneration would be very informative, especially as it relates to macrophage subtypes, re-epithelialization, cell proliferation and healing outcomes. There are several technical limitations we are trying to surmount in order to conduct this type of analysis in spiny mice and so at present we cannot provide the requested information for inclusion in this manuscript. The size of the ear pinna, and specifically the very small distance between the central (distal) part of the injury relative to the cut cartilage, make it extremely difficult to physically separate different areas within the tissue from early time points for downstream proteomic analysis. For instance, because we see CD86+ (M1) macrophages at the wound margin, but not in the blastema, we would have to ensure that proximal tissue was not included during tissue collection to correlate macrophage subtype and growth factor localization. Immunohistochemistry is the most viable option, and we are currently validating markers to downstream targets of these growth factor signaling pathways. Laser capture techniques might be possible in combination with proteomics, but this approach is beyond the scope of the current study.

5) The work of Maximinia Yun should be mentioned here. She has shown that macrophages in regenerating urodele limbs remove senescent cells and that this is obligatory for regeneration. The accumulation of senescent cells has also been implicated in the aging of organs; clearance of these cells preserves function (Baker et al., 2016). Removal of senescent cells could be a factor in Acomys ear tissue regeneration.

We thank the reviewers for drawing a very exciting connection between the role of cellular senescence and tissue regeneration. In response to this request we have amended our Discussion to include a possible role for senescent cells during regeneration based partly on Dr. Yun’s work in salamanders and newts, and on data from other regeneration models. Senescent cells could also have a positive role depending on their phenotype and we include this in our amended discussion as well. Indeed, injury-induced induction of cellular senescence has been observed during limb regeneration in salamanders and newts (Yun et al., 2015, *eLife*) and preliminary work from our lab suggests a similar induction during regeneration in spiny mice *and* scarring in lab mice. Interestingly, injury induced cellular senescence and subsequent clearance by macrophages or natural killer cells occurs during normal wound repair when a scar forms (e.g., Kang et al., 2011, Nature,Krizhanovsky et al., 2008, Cell, etc.).In these cases clearance of senescent cells appears to limit fibrosis, but it does not induce regeneration. Unfortunately, Dr. Yun’s study was not able to directly test whether the persistence of senescent cells (or an increase in their number) could inhibit regeneration due to a lack of tools in the salamander model (e.g., depleting macrophages inhibits blastema formation). Thus, her work leaves open the possibility that the type of senescent cells present upon injury might have positive or negative effects. All told, this is an active area of research for many labs and we agree with the reviewers that removal of senescent cells could be an important factor in *Acomys* ear regeneration. Furthermore, that macrophages can remove senescent cells points to an intriguing role for macrophages during regeneration.